# The Role of gp91phox and the Effect of Tranexamic Acid Administration on Hair Color in Mice

**DOI:** 10.3390/ijms20112665

**Published:** 2019-05-30

**Authors:** Keiichi Hiramoto, Yurika Yamate, Yutaka Takishita, Eisuke F. Sato

**Affiliations:** Department of Pharmaceutical Sciences, Suzuka University of Medical Science, 3500-3 Minamitamagakicho, Suzuka Mie 513-8670, Japan; m8645767@msic.med.osaka-cu.ac.jp (Y.Y.); dp16002@st.suzuka-u.ac.jp (Y.T.); efsato@suzuka-u.ac.jp (E.F.S.)

**Keywords:** tranexamic acid, Gp91phox, interleukin-1β, transforming growth factor-β, Mahogunin ring finger protein 1, Collagen XVII

## Abstract

We observed that on long-term breeding, gp91phox-knockout (gp91phox^−/−^) mice developed white hair. Here, we investigate the origin of this hitherto unexplained phenomenon. Moreover, we investigated the effect of tranexamic acid administration on the hair color in gp91phox^−/−^ mice. We administered tranexamic acid (about 12 mg/kg/day) orally to 9-week-old C57BL/6j (control) and gp91phox^−/−^ mice, thrice a week for 12 months. Compared to control mice, gp91phox^−/−^ mice showed more white hair. However, the concentrations of reactive oxygen species and the levels of interleukin (IL)-1β and transforming growth factor (TGF)-β in the skin were lower than those in the control group. Furthermore, increase in white hair was observed in the control mice upon administration of the IL-1β antagonist. On the other hand, administration of tranexamic acid led to brown colored hair on gp91phox^−/−^ mice. Although tranexamic acid treatment did not alter the expression levels of melanocortin receptor 1 and agouti signaling protein on hair follicles, it increased the expression of mahogunin ring finger protein 1 (MGRN1) and collagen XVII. These results suggested that retention of black hair requires the gp91phox/ROS/IL-1β/TGF-β pathway and that elevated levels of MGRN1 and collagen XVII lead to brown hair in gp91phox^−/−^ mice.

## 1. Introduction

The color of hair is due to melanin and is dependent on whether it is delivered to the hair matrix cells [1]. If melanin is not produced, the hair turns pale and it leads to white hair [2]. The causes of white hair include genetic factors [3], physical stress [4], under-nutrition [4], and aging [5]. With aging, the cellular activity decreases and melanin synthesis also fails at the same time. Recently, the biochemical mechanism of white hair caused by aging is clarified. The melanocyte stem cell that determines hair color, which was discovered in 2002 [1], was shown to decreased with aging, resulting in the induction of white hair [1]. Furthermore, the accumulation of DNA damage by the extrinsic factors, such as for instance, radiations, or intrinsic factors, such as active oxygen along with faulty gene restoration process could lead to depletion of melanocyte stem cells [6]. Accordingly, stopping a self-replication by serious damage leads to a depletion of a melanocyte stem cell [7]. In addition, it is reported that collagen type XVII is indispensable to a stem-cell retention system. Collagen type XVII decreases with aging, causing the melanocyte stem cells to diminish in number [8].

However, tranexamic acid (trans-4-aminomethylcyclohexanecarboxylic acid) has an antiplasmin function and inhibits the formation of linkage between plasmin and fibrin [9]. Tranexamic acid has an ameliorative effect on chloasma. Chloasma is caused by activation of plasmin and sthenia of melanogenesis and the subsequent inflammatory reaction of a dermis [10]. Tranexamic acid inhibits melanogenesis through an antiplasmin function, and at the same time it inhibits the synthesis of melanin directly in melanocytes. Tranexamic acid also performs anti-inflammatory properties [10]. Furthermore, tranexamic acid inhibits pigmentation caused by ultraviolet irradiation [11,12]. Thus, tranexamic acid has a whitening effect on the skin. However, the effect in of tranexamic acid the color of hair is not known.

In this study, we examined the influence of inflamm-aging exerts on hair color. Particularly, we investigated the role of Gp91phox, which is closely associated with inflammation and the effect of oral administration of tranexamic acid on hair color using the gp91phox-knockout (gp91phox^−/−^) mice.

## 2. Results

### 2.1. Color of Hair on the Skin of gp91phox^−/−^ mice

Although the hair color of 12-month-old C57BL/6j (control) mice was black, white hairs blend with the hair on the skin of gp91phox^−/−^ mice (Figure 1A). Moreover, ROS (Figure 1B), IL-1β (Figure 1C), and TGF-β (Figure 1D) levels in the skin of gp91phox^−/−^ mice were lower than those in the control mice. In addition, the expression of TGF-β on the niche in hair follicle of gp91phox^−/−^ mice was lower than those in the control mice (Figure 1E).

### 2.2. Effect of IL-1 Receptor Antagonist (IL-1RA) Treatment on the Hair Color in gp91phox^+/+^ Mice

The hair color of 12-month-old control mice (gp91phox^+/+^ mice) was black, however, white hair was observed after treatment with IL-1RA (an antagonist of IL-1 receptor) (Figure 2A). Moreover, the treatment of IL-1RA caused a reduction in the expression of TGF-β in the skin and the niche of hair follicles, evident from a comparison with those levels in the control group (Figure 2B,C).

### 2.3. Effect of Tranexamic Acid Treatment on the Color of Hair on the Skin

On the 12-month-old control mice, hair color was black; however, white hairs increased in number on gp91phox^−/−^ mice. On the other hand, the hair color turned brown upon oral administration of tranexamic acid (Figure 3A). The expression of MC1R and ASIP in the hair follicle did not seem differ between gp91phox^+/+^ and gp91phox^−/−^, or between tranexamic acid treated and non-treated gp91phox^−/−^ mice (Figure 3B,C). On the other hand, the expression in the hair follicle of MGRN1 and collagen XVII decreased in gp91phox^−/−^ mice, and the levels increased upon administration of tranexamic acid (Figure 3D,E).

### 2.4. Effect of Plasmin Inhibitor (Aprotinin) Treatment on the Hair Color in gp91phox^−/−^ Mice

The hair color of 12-month-old control mice was black, and white hair increased in gp91phox^−/−^ mice. The hair color became brown upon treatment with aprotinin (plasmin inhibitor) (Figure 4A). The change in hair color was prevented by aprotinin in p91phox^+/+^ mice (data not shown). The expression of MGRN1 and collagen XVII in the hair follicle decreased in gp91phox^−/−^ mice. On the aprotinin-administered gp91phox^−/−^ mice, the expression of MGRN1 was found to be elevated compared to those in the control mice (Figure 4B). On the other hand, the expression of collagen XVII was more than that in gp91phox^−/−^ mice, similar to the control mice (Figure 4C).

## 3. Discussion

In this study, under the usual rearing conditions, 12-month-old gp91phox^−/−^ mice induced white hair. In these gp91phox^−/−^ mice, the level of ROS and the expression of IL-1β and TGF-β decreased. In addition, white hair was observed in the C57BL/6j (control) mice treated with antagonist of IL-1R. On the other hand, in the gp91phox^−/−^ mice that were subjected to prolonged administration of tranexamic acid, their hair color turned brown.

In the hair follicles, the hair follicle stem cells divide in a process that eventually leads to formation of hair. A hair follicle repeats the anagen-catagen-telogen cycle, termed hair cycle, and activation of hair follicle stem cells in an anagen phase is a crucial step for the production of hair [13]. In pigment cells, melanin is produced within a melanosome, which is supplied to the keratinocyte of epidermis or hair follicles inducing pigmentation in hair. These pigment cells are not maintained throughout life but continue to be renewed. Existence of a melanocyte stem cell was found to be the source of pigment cells [1]. TGF-β is concerned with the retention of these melanocyte stem cells [6]. In gp91phox^−/−^ mice used in this experiment, expression of TGF-β was found to be decreasing, ranging from the bulge area to the sub-bulge area where a hair follicle stem cells occur. Perhaps due to this, retention of the melanocyte stem cells was not achieved, and their numbers diminished continually, leading to the growth of white hair. The role of Gp91phox NADPH oxidase enzyme in generalizing ROS is known [14]. With a decrease in TGF-β, the control mechanism of the ROS generation by NOx may be affected. Thus, we examined the expression of the nucleotide-binding domain, the leucine-rich-containing family, pyrin domain-containing-3 (NLRP3) and caspase-1 in the mouse skin. An increase in the expression of NLRP3 and caspase-1 was seen following the certain reason, the gp91phox^−/−^ mice did not show an increased expression. Furthermore, the skin IL-1β level increased in the control mice, but in gp91phox^−/−^ mice, the expression of IL-1β was hardly observed. IL-1β induced expression of TGF-β [15,16]. Furthermore, decrease of IL-1β might inhibit expression of TGF-β in the bulge region, and may have caused the number of melanocyte stem cells to decrease. We then treated the control mice with the antagonist of IL-1β receptor over a long period of time, following which white hair-expression was observed, corroborating with our previous results. On the other hand, IL-1β is secreted from various cells of skin, such as melanocytes and keratinocytes [17]. This study focuses on hair color, and the IL-1β secreted from melanocytes could play an important role in hair follicles. Therefore, further studies are necessary for the involvement of melanocytes and IL-1β.

The hair color of gp91phox^-/-^ mice turned brown upon administration of tranexamic acid. Generally, eumelanin is dominant in black hair, and pheomelanin is dominant in brown hair [18]. When a α-melanocyte stimulating hormone (α-MSH) combines with MC1R, cAMP increases, an activation of microphthalmia-associated transcription factor (MITF) occurs and, as a result, eumelanin is made [19]. Moreover, cAMP inhibits the synthesis of MGRN1, which inhibits the gene responsible for MITF synthetic pathway [20]. On the other hand, ASIP combines with MC1R and obstructs the activity of cAMP; as a result, the signal that leads to production of eumelanin stops [19]. At the same time, ASIP/MC1R obstructs a specific gene and induces the synthesis of MGRN1, which in turn suppresses the synthesis of MITF leading to decreased levels of eumelanin.

In this study, we observed that the expression of MC1R and ASIP did not change upon administration of tranexamic acid to the gp91phox^−/−^ mice. However, the increase in MGRN1 was observed. From these results, tranexamic acid may have affected the gene, which is concerned with the control of MGRN1, downstream to α-MSH/MC1R or ASIP/MC1R. The effect of tranexamic acid is that of a plasmin inhibitor, as described previously. Therefore, tranexamic acid inhibits a plasmin, causing MGRN1 and pheomelanin to increase. Thus, tranexamic acid was thought to turn the color brown. Consequently, the gene which controls MGRN1 could be a plasmin or plasmin associated gene.

However, it is reported that plasmin cuts the extracellular region of the collagen XVII in an NC16A domain [21]. With the loss of collagen XVII, the stemness of a hair follicle stem cell is also lost, hence the stem cell changes to the epidermal keratinocyte [7]. The effect of a plasmin is inhibited by tranexamic acid administration and loss of collagen XVII is suppressed. As a result, tranexamic acid may be activating the differentiation to a pigmentation stem cell and pheomelanin synthesis.

Therefore, in order to validate our hypothesis, we conducted the experiment with aprotinin, which should inhibit plasmin except in tranexamic acid administered gp91phox^−/−^ mice. Aprotinin is serine protease inhibitor while tranexamic acid is a lysine analog [22]. The hair color of the gp91phox^−/−^ mice turned brown through aprotinin administration (Figure 4A). In addition, the expression of MGRN1 in hair follicle increased (Figure 4B). From these findings, we may infer that plasmin is closely related to expression of MGRN1. Furthermore, the expression of the collagen XVII was found to be increased by aprotinin.

In this study, the black hair maintenance is better with the ROS (gp91phox^+/+^ mice) rather than gp91phox^−/−^ mice; it is different to the usual aging state, where high ROS causes white hair. The 12-month-old mice which we used in this experiment could not be said to be aging mice, as it is not clear that this phenomenon is associated to aging. However, in humans, if the decrease of gp91phox occurs, this phenomenon is expressed. On the other hand, tranexamic acid has the effect of anti-chronic inflammation, which is the factor of white hair. Thus, it is possible that tranexamic acid administration influences hair color caused by aging.

From the above results, we considered the reason for white hair in gp91phox^−/−^ mice to be the suppression of the ROS/IL-1β/TGF-β signaling pathway. Moreover, increase of MGRN1 and loss of inhibition of collagen XVII were considered to be the causes for the hair to turn brown upon tranexamic acid administration. However, the detailed mechanisms behind these phenomena are not known. Therefore, the further studies are necessary.

## 4. Materials and Methods

### 4.1. Ethical Approval

This study was carried out in strict accordance with the recommendations in the Guide for the Care and Use of Laboratory Animals of Suzuka University of Medical Science (Approval number: 34, date of acquisition: 01 July 2015). All surgery was performed under sodium pentobarbital anesthesia, and all efforts were made to minimize suffering.

### 4.2. Animal Experiments

Nine-week-old male C57BL/6j mice (SLC, Hamamatsu, Shizuoka, Japan) and gp91phox^-/-^ mice (Jackson Laboratories, Bar Harbour, ME, USA) were used for the experiments. The mice were kept on a 12-h light/12-h dark cycle at 23 ± 1 °C under specific-pathogen-free (SPF) conditions, and all animals were allowed free access to laboratory chow (rodent diet EQ 5L37; SLC, Hamamatsu, Shizuoka, Japan) and water during the experiments. The animal was not exposed to ultraviolet rays, and care was taken ensure no physical stress load at the start. We used ten per group and repeated the experiments three times. This study was carried out in strict accordance with the recommendations of the Guide for the Care and Use of Laboratory Animals of the Suzuka University of Medical Science (approval number: 34). All surgeries were performed under pentobarbital anesthesia, and efforts were made to minimize animal suffering.

### 4.3. Tranexamic Acid Treatment

Approximately 12 mg/kg of tranexamic acid (Daiichi Sankyo Healthcare Co., Ltd., Tokyo, Japan) in distilled water was orally administered to mice three times per week for 1 year. The solvent-administered animals were administered distilled water [11].

### 4.4. Interleukin 1 Receptor Antagonist (IL-1RA) Treatment

Approximately 10 μg/100 μL of IL-1RA (ATGen Co., Ltd., Gyeonggi-do, Korea) in saline was intraperitoneally injected into C57BL/6j mice three times per week for 1 year. The solvent-administered animals were administered saline [23,24].

### 4.5. Plasmin Inhibitor (Aprotinin) Treatment

Approximately 100 KIU/kg of plasmin inhibitor (Cayman, Ann Arbor, MI, USA) in saline was intraperitoneally injected into gp91phox^−/−^ mice three times per week for 1 year. The solvent-administered animals were administered saline [25,26].

### 4.6. Preparation and Staining of the Dorsal Skin and Hair Follicle

We obtained dorsal skin and hair follicle samples 1-year after the start of the experiment. The specimens were fixed in phosphate-buffered paraformaldehyde (4%), embedded in frozen Tissue Tek, an Optimal-Cutting Temperature (OCT) compound, and cut into 5-μm sections. The sections were subjected to immunostaining. We described the details of the staining method in our previous study [27]. Briefly, the specimens were incubated with rabbit polyclonal anti-mahogunin ring finger protein 1 (MGRN1) (1:100; Abnova, Taipei City, Taiwan), rabbit monoclonal anti collagen XVII (1:100; Abcam, Cambridge, MA, USA), rabbit polyclonal anti-agouti signaling protein (ASIP) (1:100; GeneTex, Irvine, CA, USA), rabbit polyclonal anti melanocortin receptor 1 (MC1R) (1:100; GeneTex) or rabbit polyclonal anti transforming growth factor (TGF)-β (1:100; Abcam) primary antibodies. The samples were then washed and incubated with fluorescein isothiocyanate-conjugated anti-rabbit and tetramethylrhodamine isothiocyanate-conjugated anti-rabbit (1:30; Dako Cytomation, Glstrup, Denmark) secondary antibodies. The expression levels were evaluated immunohistochemically using a fluorescence microscope.

### 4.7. Measurement of Skin on IL-1β, TGF-β and Reactive Oxygen Species (ROS) Levels

Skin samples were collected the last day of the experiment. The skin level of IL-1βand TGF-β was determined using commercial enzyme-linked immunosorbent assay (ELISA) kits (IL-1β; Abcam; TGF-β; MyBioSource, SanDiego, CA, USA) according to the instruction of the manufacturer. We also measured the skin ROS concentration using an assay kit (OxiSelect^TM^ In Vitro ROS/RNA Assaykit, STA-347; Cell Biolabs, INC., San Diego, CA, USA) according to the instruction of the manufacturer.

### 4.8. Statistical Analyses

All data are presented as means ± standard deviation (SD). For comparisons among groups, Student’s *t*-test was applied, with *p* < 0.05 considered to be statistically significant.

## 5. Conclusions

In this study, white hair grew on gp91phox^−/−^ mice, and when the prolonged administration of the tranexamic acid was carried out to this gp91phox^−/−^ mice, their hair turned brown. Our results suggest that gp91phox is concerned with TGF-β of the niche of the hair follicle, and also that plasmin modulates MGRN1 and collagen XVII (Figure 5). These findings may lead to an amelioration of psilosis or white hair and may improve the quality of lives of many.

## Figures and Tables

**Figure 1 ijms-20-02665-f001:**
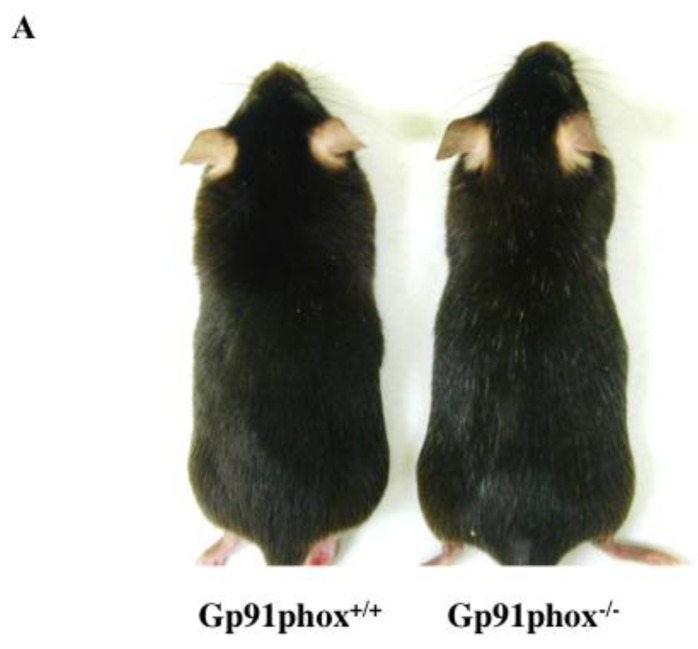
The color of hair on the dorsal skin in twelve-months-old gp91phox^−/−^ mice (**A**). An analysis of the skin ROS (**B**), IL-1β (**C**), and TGF-β (**D**) concentrations in twelve-month-old gp91phox^−/−^ mice. The expression levels of TGF-β in the niche of hair follicle in dorsal skin of the gp91phox^−/−^ mice at twelve-months old (**E**). The data show the results from one typical experiment involving six animals. The values are expressed as mean ± standard deviation (SD) derived from six animals. * *p* < 0.05. Scale bar = 10 μm.

**Figure 2 ijms-20-02665-f002:**
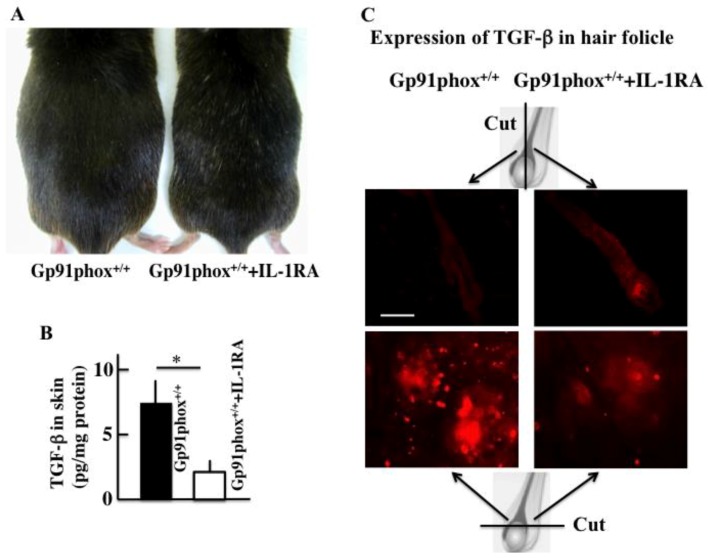
The effect on IL-1βRA treatment on the hair color in twelve-month-old C57BL/6j mice (**A**). The effect of IL-1βRA treatments on the plasma level TGF-β in the gp91phox^−/−^ mice (**B**) and the expression of TGF-β in the niche of hair follicle in the dorsal skin (**C**). The data show the results from one typical experiment involving six animals. The values are expressed as mean ± SD derived from six animals. * *p* < 0.05. Scale bar = 10 μm.

**Figure 3 ijms-20-02665-f003:**
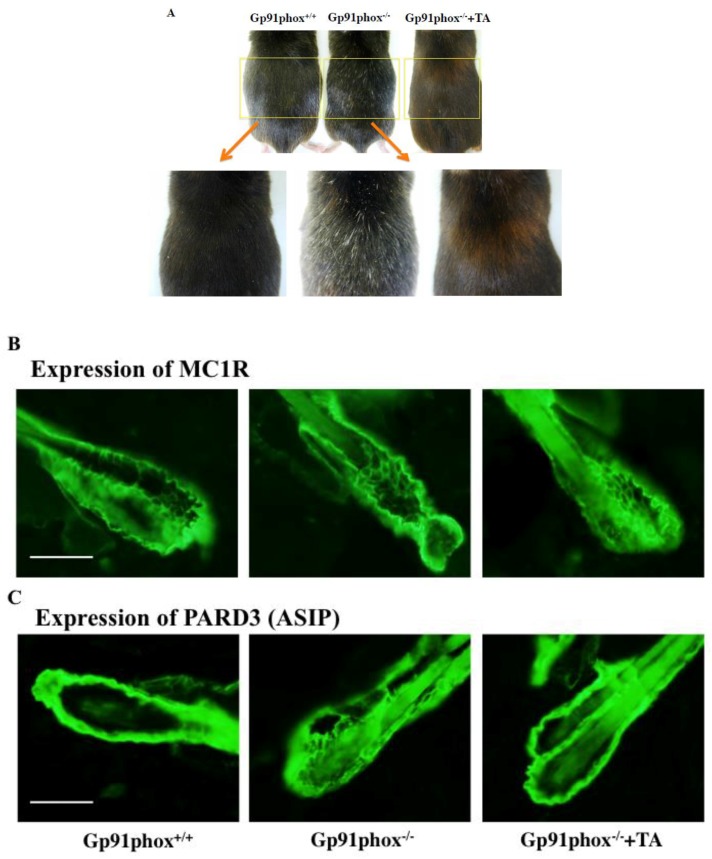
The effect of tranexamic acid on the hair color (**A**) and the effect of tranexamic acid on the expression of MC1R (**B**), ASIP (**C**), MGRN1 (**D**), and collagen XVII (**E**) in the twelve-month-old gp91phox^−/−^ mice. The data show the results from one typical experiment involving six animals. Scale bar = 10 μm.

**Figure 4 ijms-20-02665-f004:**
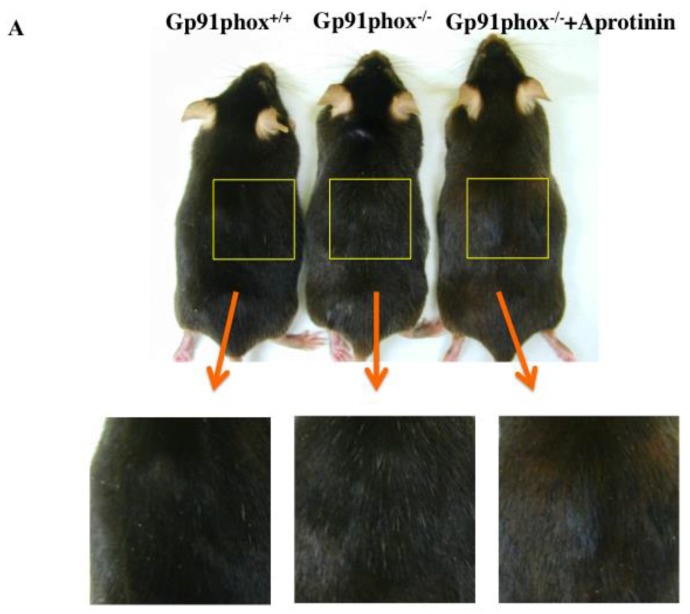
The effect of aprotinin on the hair color (**A**) and the effect of tranexamic acid on the expression of MGRN1 (**B**) and collagen XVII (**C**) in the twelve-months-old gp91phox^−/−^ mice. The data show the results from one typical experiment involving six animals. Scale bar = 10 μm.

**Figure 5 ijms-20-02665-f005:**
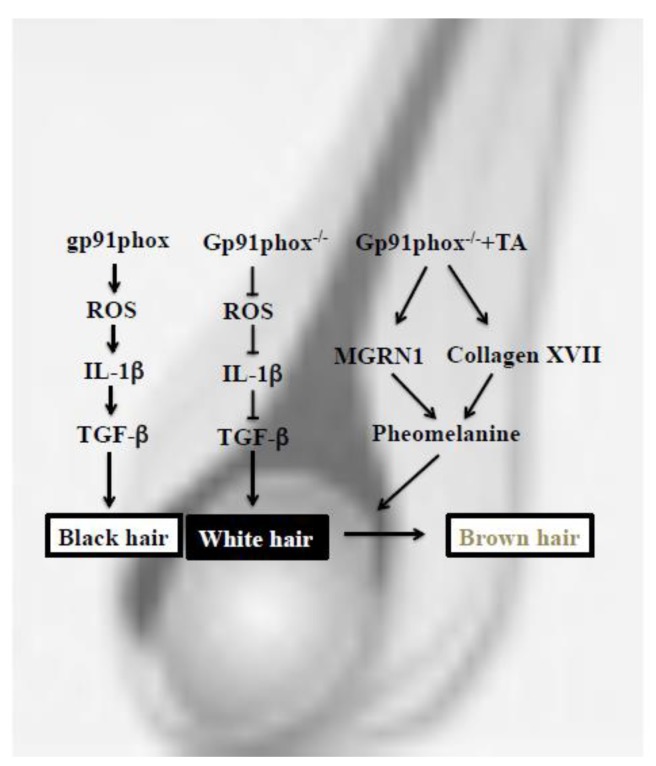
Mechanism of the effect of tranexamic acid on gp91phox^−/−^ mice.

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
