# Peer review of "The Role of gp91phox and the Effect of Tranexamic Acid Administration on Hair Color in Mice"

_ijms, 2019, doi:10.3390/ijms20112665_

Round 1
Reviewer 1 Report
The paper is interesting and it should be interesting to control the activity of chitin by the same technology.Our group,in fact,many years ago verified the activity of topical administration of oral chitin on mouse with gray hairs due to a diet of low content of chitin.
Author Response
Responses to the comments of Reviewer 1:
We thank the reviewer for these insightful comments as they have helped to significantly improve the manuscript.
We also want to go further research.

Reviewer 2 Report
The authors reports the reason for the white hair of gp91phox-/- mice that is the suppression of the ROS/IL-1/TGF signaling pathway.
This is intriguing because if one considers that this phenomenon is usually associated to aging, a chronic inflammatory state has been expected (please discuss this point).
However melanocytes also have a role in immune system showing many characteristics of dendritic cells which are important antigen presenting cells. These cells can counteract stress response by producing many pro-cytokines such as IL-1beta etc. So melanocyte-secreted factors may be indispensable for the maintenance of the black and brown hair (please discuss). To this end , if reversion experiments with blocking IL-1R have been done, the authors should also test the effects of recombinant IL-1. The may give adjunct value to the presenting data.
Author Response
Responses to the comments of Reviewer 2:
We thank the reviewer for these insightful comments as they have helped to significantly improve the manuscript.
1. Thank you for your comment. We added the idea of the relationship between age-relating chronic inflammation and hair color.
(Discussion: p.7 line 47 to p. 8 line 2)
In this study, the black hair maintenances better with the ROS (gp91phox+/+ mice) rather than gp91phox-/- mice. Then it is difference to usual aging state which high ROS causes white hair. The 12-month-old mice which we used in this experiment can not be said as aging mice. Then it is not clear that this phenomenon is associated to aging. But in human, if the decrease of gp91phox may occur, this phenomenon will express. On the other hand, tranexamic acid has the effect of anti-chronic inflammation which the factor of white hair. Then it is possible that tranexamic acid administration will influence to the hair color of aging.
2. Thank you for your comment. We added to “Discussion” part about the relation between IL-1b and melanocyte.
(Discussion: p.7 lines 13 to 16)
On the other hand, IL-1b is secreted from various cells of skin such as melanocytes and keratinocytes [22]. This study focuses on hair color, the IL-1b secreted from melanocytes could play an important role in hair follicles. Therefore, further studies are necessary for the involvement of melanocytes and IL-1b.
(References:)
22. Tam, I.; Stepien, K. Secretion of proinflammatory cytokines by normal human melanocytes in response to lipopolysaccharide. Acta Biochim. Pol. 2011, 58, 507-511.
3. We thank the reviewer for this comment. The administration test of recombinant IL-1b can not be performed at present because the gp91phox-/- mice can not be supplied. We would like to implement it once we have an outlook on supply.
